# Envy-Free Classification

**Maria-Florina Balcan**
Machine Learning Department
Carnegie Mellon University
ninamf@cs.cmu.edu

**Travis Dick**
Computer Science Department
Carnegie Mellon University
tdick@cs.cmu.edu

**Ritesh Noothigattu**
Machine Learning Department
Carnegie Mellon University
riteshn@cmu.edu

**Ariel D. Procaccia**
Computer Science Department
Carnegie Mellon University
arielpro@cs.cmu.edu

## Abstract

In classic fair division problems such as cake cutting and rent division, *envy-freeness* requires that each individual (weakly) prefer his allocation to anyone else's. On a conceptual level, we argue that envy-freeness also provides a compelling notion of fairness for classification tasks, especially when individuals have heterogeneous preferences. Our technical focus is the *generalizability* of envy-free classification, i.e., understanding whether a classifier that is envy free on a sample would be almost envy free with respect to the underlying distribution with high probability. Our main result establishes that a small sample is sufficient to achieve such guarantees, when the classifier in question is a mixture of deterministic classifiers that belong to a family of low Natarajan dimension.

## 1 Introduction

The study of fairness in machine learning is driven by an abundance of examples where learning algorithms were perceived as discriminating against protected groups [29, 6]. Addressing this problem requires a conceptual — perhaps even philosophical — understanding of what fairness means in this context. In other words, the million dollar question is (arguably[1]) this: What are the formal constraints that fairness imposes on learning algorithms?

In this paper, we propose a new measure of algorithmic fairness. It draws on an extensive body of work on rigorous approaches to fairness, which — modulo one possible exception (see Section 1.2) — has not been tapped by machine learning researchers: the literature on *fair division* [3, 20]. The most prominent notion is that of *envy-freeness* [10, 31], which, in the context of the allocation of goods, requires that the utility of each individual for his allocation be at least as high as his utility for the allocation of any other individual; for six decades, it has been the gold standard of fairness for problems such as cake cutting [25, 24] and rent division [28, 12]. In the classification setting, envy-freeness would simply mean that the utility of each individual for his distribution over outcomes is at least as high as his utility for the distribution over outcomes assigned to any other individual.

It is important to say upfront that envy-freeness is *not* suitable for several widely-studied problems where there are only two possible outcomes, one of which is 'good' and the other 'bad'; examples include predicting whether an individual would default on a loan, and whether an offender would recidivate. In these degenerate cases, envy-freeness would require that the classifier assign each and every individual the exact same probability of obtaining the 'good' outcome, which, clearly, is not a reasonable constraint.

By contrast, we are interested in situations where there is a diverse set of possible outcomes, and individuals have diverse preferences for those outcomes. For example, consider a system responsible for displaying credit card advertisements to individuals. There are many credit cards with different eligibility requirements, annual rates, and reward programs. An individual's utility for seeing a card's advertisement will depend on his eligibility, his benefit from the rewards programs, and potentially other factors. It may well be the case that an envy-free advertisement assignment shows Bob advertisements for a card with worse annual rates than those shown to Alice; this outcome is not unfair if Bob is genuinely more interested in the card offered to him. Such rich utility functions are also evident in the context of job advertisements [6]: people generally want higher paying jobs, but would presumably have higher utility for seeing advertisements for jobs that better fit their qualifications and interests.

A second appealing property of envy-freeness is that its fairness guarantee binds at the level of individuals. Fairness notions can be coarsely characterized as being either individual notions, or group notions, depending on whether they provide guarantees to specific individuals, or only on average to a protected subgroup. The majority of work on fairness in machine learning focuses on group fairness [18, 9, 35, 13, 15, 34].

There is, however, one well-known example of individual fairness: the influential fair classification model of Dwork et al. [9]. The model involves a set of individuals and a set of outcomes. The centerpiece of the model is a *similarity metric* on the space of individuals; it is specific to the classification task at hand, and ideally captures the ethical ground truth about relevant attributes. For example, a man and a woman who are similar in every other way should be considered similar for the purpose of credit card offerings, but perhaps not for lingerie advertisements. Assuming such a metric is available, fairness can be naturally formalized as a Lipschitz constraint, which requires that individuals who are close according to the similarity metric be mapped to distributions over outcomes that are close according to some standard metric (such as total variation).

As attractive as this model is, it has one clear weakness from a practical viewpoint: the availability of a similarity metric. Dwork et al. [9] are well aware of this issue; they write that justifying this assumption is "one of the most challenging aspects" of their approach. They add that "in reality the metric used will most likely only be society's current best approximation to the truth." But, despite recent progress on automating ethical decisions in certain domains [23, 11], the task-specific nature of the similarity metric makes even a credible approximation thereof seem unrealistic. In particular, if one wanted to learn a similarity metric, it is unclear what type of examples a relevant dataset would consist of.

In place of a metric, envy-freeness requires access to individuals' utility functions, but — by contrast — we do not view this assumption as a barrier to implementation. Indeed, there are a variety of techniques for learning utility functions [4, 22, 2]. Moreover, in our running example of advertising, one can use standard measures like expected click-through rate (CTR) as a good proxy for utility.

It is worth noting that the classification setting is different from classic fair division problems in that the "goods" (outcomes) are non-excludable. In fact, one envy-free solution simply assigns each individual to his favorite outcome. But this solution may be severely suboptimal according to another (standard) component of our setting, the *loss function*, which, in the examples above, might represent the expected revenue from showing an ad to an individual. Typically the loss function is not perfectly aligned with individual utilities, and, therefore, it may be possible to achieve smaller loss than the naïve solution without violating the envy-freeness constraint.

In summary, we view envy-freeness as a compelling, well-established, and, importantly, practicable notion of individual fairness for classification tasks with a diverse set of outcomes when individuals have heterogeneous preferences. Our goal is to understand its learning-theoretic properties.

## 1.1 Our Results

The challenge is that the space of individuals is potentially huge, yet we seek to provide universal envy-freeness guarantees. To this end, we are given a sample consisting of individuals drawn from an unknown distribution. We are interested in learning algorithms that minimize loss, subject to satisfying the envy-freeness constraint, *on the sample*. Our primary technical question is that of generalizability, that is, *given a classifier that is envy free on a sample, is it approximately envy free on the underlying distribution?* Surprisingly, Dwork et al. [9] do not study generalizability in their

model, and we are aware of only one subsequent paper that takes a learning-theoretic viewpoint on individual fairness and gives theoretical guarantees (see Section 1.2).

In Section 3, we do not constrain the classifier. Therefore, we need some strategy to extend a classifier that is defined on a sample; assigning an individual the same outcome as his *nearest neighbor* in the sample is a popular choice. However, we show that *any* strategy for extending a classifier from a sample, on which it is envy free, to the entire set of individuals is unlikely to be approximately envy free on the distribution, unless the sample is exponentially large.

For this reason, in Section 4, we focus on structured families of classifiers. On a high level, our goal is to relate the combinatorial richness of the family to generalization guarantees. One obstacle is that standard notions of dimension do not extend to the analysis of randomized classifiers, whose range is *distributions* over outcomes (equivalently, real vectors). We circumvent this obstacle by considering mixtures of *deterministic* classifiers that belong to a family of bounded Natarajan dimension (an extension of the well-known VC dimension to multi-class classification). Our main theoretical result asserts that, under this assumption, envy-freeness on a sample does generalize to the underlying distribution, even if the sample is relatively small (its size grows almost linearly in the Natarajan dimension).

Finally, in Section 5, we design and implement an algorithm that learns (almost) envy-free mixtures of linear one-vs-all classifiers. We present empirical results that validate our computational approach, and indicate good generalization properties even when the sample size is small.

## 1.2 Related Work

Conceptually, our work is most closely related to work by Zafar et al. [34]. They are interested in group notions of fairness, and advocate preference-based notions instead of parity-based notions. In particular, they assume that each group has a utility function for *classifiers*, and define the *preferred treatment* property, which requires that the utility of each group for its own classifier be at least its utility for the classifier assigned to any other group. Their model and results focus on the case of binary classification where there is a desirable outcome and an undesirable outcome, so the utility of a group for a classifier is simply the fraction of its members that are mapped to the desirable outcome. Although, at first glance, this notion seems similar to envy-freeness, it is actually fundamentally different.[2] Our paper is also completely different from that of Zafar et al. in terms of technical results; theirs are purely empirical in nature, and focus on the increase in accuracy obtained when parity-based notions of fairness are replaced with preference-based ones.

Concurrent work by Rothblum and Yona [26] provides generalization guarantees for the metric notion of individual fairness introduced by Dwork et al. [9], or, more precisely, for an approximate version thereof. There are two main differences compared to our work: first, we propose envy-freeness as an alternative notion of fairness that circumvents the need for a similarity metric. Second, they focus on randomized *binary* classification, which amounts to learning a real-valued function, and so are able to make use of standard Rademacher complexity results to show generalization. By contrast, standard tools do not directly apply in our setting. It is worth noting that several other papers provide generalization guarantees for notions of group fairness, but these are more distantly related to our work [35, 32, 8, 16, 14].

## 2 The Model

We assume that there is a space $\mathcal{X}$ of individuals, a finite space $\mathcal{Y}$ of outcomes, and a utility function $u : \mathcal{X} \times \mathcal{Y} \to [0, 1]$ encoding the preferences of each individual for the outcomes in $\mathcal{Y}$. In the advertising example, individuals are users, outcomes are advertisements, and the utility function reflects the benefit an individual derives from being shown a particular advertisement. For any distribution $p \in \Delta(\mathcal{Y})$ (where $\Delta(\mathcal{Y})$ is the set of distributions over $\mathcal{Y}$) we let $u(x, p) = \mathbb{E}_{y \sim p}[u(x, y)]$ denote individual $x$'s expected utility for an outcome sampled from $p$. We refer to a function $h : \mathcal{X} \to \Delta(\mathcal{Y})$ as a *classifier*, even though it can return a distribution over outcomes.

## 2.1 Envy-Freeness

Roughly speaking, a classifier $h : \mathcal{X} \to \Delta(\mathcal{Y})$ is envy free if no individual prefers the outcome distribution of someone else over his own.

**Definition 1.** A classifier $h : \mathcal{X} \to \Delta(\mathcal{Y})$ is *envy free (EF)* on a set $S$ of individuals if $u(x, h(x)) \geq u(x, h(x'))$ for all $x, x' \in S$. Similarly, $h$ is $(\alpha, \beta)$-*EF* with respect to a distribution $P$ on $\mathcal{X}$ if

$$\Pr_{x,x' \sim P}\big(u(x, h(x)) < u(x, h(x')) - \beta\big) \leq \alpha.$$

Finally, $h$ is $(\alpha, \beta)$-*pairwise EF* on a set of pairs of individuals $S = \{(x_i, x_i')\}_{i=1}^n$ if

$$\frac{1}{n} \sum_{i=1}^n \mathbb{I}\{u(x_i, h(x_i)) < u(x_i, h(x_i')) - \beta\} \leq \alpha.$$

Any classifier that is EF on a sample $S$ of individuals is also $(\alpha, \beta)$-pairwise EF on any pairing of the individuals in $S$, for any $\alpha \geq 0$ and $\beta \geq 0$. The weaker pairwise EF condition is all that is required for our generalization guarantees to hold.

## 2.2 Optimization and Learning

Our formal learning problem can be stated as follows. Given sample access to an unknown distribution $P$ over individuals $\mathcal{X}$ and their utility functions, and a known loss function $\ell : \mathcal{X} \times \mathcal{Y} \to [0, 1]$, find a classifier $h : \mathcal{X} \to \Delta(\mathcal{Y})$ that is $(\alpha, \beta)$-EF with respect to $P$ minimizing expected loss $\mathbb{E}_{x \sim P}[\ell(x, h(x))]$, where for $x \in \mathcal{X}$ and $p \in \Delta(\mathcal{Y})$, $\ell(x, p) = \mathbb{E}_{y \sim p}[\ell(x, y)]$.

We follow the empirical risk minimization (ERM) learning approach, i.e., we collect a sample of individuals drawn i.i.d from $P$ and find an EF classifier with low loss on the sample. Formally, given a sample of individuals $S = \{x_1, \ldots, x_n\}$ and their utility functions $u_{x_i}(\cdot) = u(x_i, \cdot)$, we are interested in a classifier $h : S \to \Delta(\mathcal{Y})$ that minimizes $\sum_{i=1}^n \ell(x_i, h(x_i))$ among all classifiers that are EF on $S$.

Recall that we consider randomized classifiers that can assign a distribution over outcomes to each of the individuals. However, one might wonder whether the EF classifier that minimizes loss on a sample happens to always be deterministic. Or, at least, the optimal deterministic classifier on the sample might incur a loss that is very close to that of the optimal randomized classifier. If this were true, we could restrict ourselves to classifiers of the form $h : \mathcal{X} \to \mathcal{Y}$, which would be much easier to analyze. Unfortunately, it turns out that this is not the case. In fact, there could be an arbitrary (multiplicative) gap between the optimal randomized EF classifier and the optimal deterministic EF classifier. The intuition behind this is as follows. A deterministic classifier that has very low loss on the sample, but is not EF, would be completely discarded in the deterministic setting. On the other hand, a randomized classifier could take this loss-minimizing deterministic classifier and mix it with a classifier with high "negative envy", so that the mixture ends up being EF and at the same time has low loss. This is made concrete in the following example.

**Example 1.** Let $S = \{x_1, x_2\}$ and $\mathcal{Y} = \{y_1, y_2, y_3\}$. Let the loss function be such that

$$\begin{aligned}
\ell(x_1, y_1) = 0 \qquad & \ell(x_1, y_2) = 1 \qquad && \ell(x_1, y_3) = 1 \\
\ell(x_2, y_1) = 1 \qquad & \ell(x_2, y_2) = 1 \qquad && \ell(x_2, y_3) = 0
\end{aligned}$$

Moreover, let the utility function be such that

$$\begin{aligned}
u(x_1, y_1) = 0 \qquad & u(x_1, y_2) = 1 \qquad && u(x_1, y_3) = \frac{1}{\gamma} \\
u(x_2, y_1) = 0 \qquad & u(x_2, y_2) = 0 \qquad && u(x_2, y_3) = 1
\end{aligned}$$

where $\gamma > 1$. The only deterministic classifier with a loss of 0 is $h_0$ such that $h_0(x_1) = y_1$ and $h_0(x_2) = y_3$. But, this is not EF, since $u(x_1, y_1) < u(x_1, y_3)$. Furthermore, every other deterministic classifier has a total loss of at least 1, causing the optimal deterministic EF classifier to have loss of at least 1.

To show that randomized classifiers can do much better, consider the randomized classifier $h_*$ such that $h_*(x_1) = (1 - 1/\gamma, 1/\gamma, 0)$ and $h_*(x_2) = (0, 0, 1)$. This classifier can be seen as a mixture of

the classifier $h_0$ of 0 loss, and the deterministic classifier $h_e$, where $h_e(x_1) = y_2$ and $h_e(x_2) = y_3$, which has high "negative envy". One can observe that this classifier $h_*$ is EF, and has a loss of just $1/\gamma$. Hence, the loss of the optimal randomized EF classifier is $\gamma$ times smaller than the loss of the optimal deterministic one, for any $\gamma > 1$.

## 3 Arbitrary Classifiers

An important (and typical) aspect of our learning problem is that the classifier $h$ needs to provide an outcome distribution for every individual, not just those in the sample. For example, if $h$ chooses advertisements for visitors of a website, the classifier should still apply when a new visitor arrives. Moreover, when we use the classifier for new individuals, it must continue to be EF. In this section, we consider two-stage approaches that first choose outcome distributions for the individuals in the sample, and then extend those decisions to the rest of $\mathcal{X}$.

In more detail, we are given a sample $S = \{x_1, \ldots, x_n\}$ of individuals and a classifier $h : S \to \Delta(\mathcal{Y})$ assigning outcome distributions to each individual. Our goal is to extend these assignments to a classifier $\overline{h} : \mathcal{X} \to \Delta(\mathcal{Y})$ that can be applied to new individuals as well. For example, $h$ could be the loss-minimizing EF classifier on the sample $S$.

For this section, we assume that $\mathcal{X}$ is equipped with a distance metric $d$. Moreover, we assume in this section that the utility function $u$ is $L$-Lipschitz on $\mathcal{X}$. That is, for every $y \in \mathcal{Y}$ and for all $x, x' \in \mathcal{X}$, we have $|u(x, y) - u(x', y)| \le L \cdot d(x, x')$.

Under the foregoing assumptions, one natural way to extend the classifier on the sample to all of $\mathcal{X}$ is to assign new individuals the same outcome distribution as their nearest neighbor in the sample. Formally, for a set $S \subset \mathcal{X}$ and any individual $x \in \mathcal{X}$, let $\mathrm{NN}_S(x) \in \arg\min_{x' \in S} d(x, x')$ denote the nearest neighbor of $x$ in $S$ with respect to the metric $d$ (breaking ties arbitrarily). The following simple result (whose proof is relegated to Appendix B) establishes that this approach preserves envy-freeness in cases where the sample is exponentially large.

**Theorem 1.** *Let $d$ be a metric on $\mathcal{X}$, $P$ be a distribution on $\mathcal{X}$, and $u$ be an $L$-Lipschitz utility function. Let $S$ be a set of individuals such that there exists $\hat{\mathcal{X}} \subset \mathcal{X}$ with $P(\hat{\mathcal{X}}) \ge 1 - \alpha$ and $\sup_{x \in \hat{\mathcal{X}}} d(x, \mathrm{NN}_S(x)) \le \beta/(2L)$. Then for any classifier $h : S \to \Delta(\mathcal{Y})$ that is EF on $S$, the extension $\overline{h} : \mathcal{X} \to \Delta(\mathcal{Y})$ given by $\overline{h}(x) = h(\mathrm{NN}_S(x))$ is $(\alpha, \beta)$-EF on $P$.*

The conditions of Theorem 1 require that the set of individuals $S$ is a $\beta/(2L)$-net for at least a $(1-\alpha)$-fraction of the mass of $P$ on $\mathcal{X}$. In several natural situations, an exponentially large sample guarantees that this occurs with high probability. For example, if $\mathcal{X}$ is a subset of $\mathbb{R}^q$, $d(x, x') = \|x - x'\|_2$, and $\mathcal{X}$ has diameter at most $D$, then for any distribution $P$ on $\mathcal{X}$, if $S$ is an i.i.d. sample of size $O(\frac{1}{\alpha}(\frac{LD\sqrt{q}}{\beta})^q(q \log \frac{LD\sqrt{q}}{\beta} + \log \frac{1}{\delta}))$, it will satisfy the conditions of Theorem 1 with probability at least $1 - \delta$. This sampling result is folklore, but, for the sake of completeness, we prove it in Lemma 3 of Appendix B.

However, the exponential upper bound given by the nearest neighbor strategy is as far as we can go in terms of generalizing envy-freeness from a sample (without further assumptions). Specifically, our next result establishes that *any* algorithm — even randomized — for extending classifiers from the sample to the entire space $\mathcal{X}$ requires an exponentially large sample of individuals to ensure envy-freeness on the distribution $P$. The proof of Theorem 2 can be found in Appendix B.

**Theorem 2.** *There exists a space of individuals $\mathcal{X} \subset \mathbb{R}^q$, and a distribution $P$ over $\mathcal{X}$ such that, for every randomized algorithm $\mathcal{A}$ that extends classifiers on a sample to $\mathcal{X}$, there exists an $L$-Lipschitz utility function $u$ such that, when a sample of individuals $S$ of size $n = 4^q/2$ is drawn from $P$ without replacement, there exists an EF classifier on $S$ for which, with probability at least $1 - 2\exp(-4^q/100) - \exp(-4^q/200)$ jointly over the randomness of $\mathcal{A}$ and $S$, its extension by $\mathcal{A}$ is not $(\alpha, \beta)$-EF with respect to $P$ for any $\alpha < 1/25$ and $\beta < L/8$.*

We remark that a similar result would hold even if we sampled $S$ with replacement; we sample here without replacement purely for ease of exposition.

# 4 Low-Complexity Families of Classifiers

In this section we show that (despite Theorem 2) generalization for envy-freeness is possible using much smaller samples of individuals, as long as we restrict ourselves to classifiers from a family of relatively low complexity.

In more detail, two classic complexity measures are the VC-dimension [30] for binary classifiers, and the Natarajan dimension [21] for multi-class classifiers. However, to the best of our knowledge, there is no suitable dimension directly applicable to functions ranging over distributions, which in our case can be seen as $|\mathcal{Y}|$-dimensional real vectors. One possibility would be to restrict ourselves to deterministic classifiers of the type $h : \mathcal{X} \to \mathcal{Y}$, but we have seen in Section 2 that envy-freeness is a very strong constraint on deterministic classifiers. Instead, we will consider a family $\mathcal{H}$ consisting of randomized mixtures of $m$ deterministic classifiers belonging to a family $\mathcal{G} \subset \{g : \mathcal{X} \to \mathcal{Y}\}$ of low Natarajan dimension. This allows us to adapt Natarajan-dimension-based generalization results to our setting while still working with randomized classifiers. The definition and relevant properties of the Natarajan dimension are summarized in Appendix A.

Formally, let $\vec{g} = (g_1, \ldots, g_m) \in \mathcal{G}^m$ be a vector of $m$ functions in $\mathcal{G}$ and $\eta \in \Delta_m$ be a distribution over $[m]$, where $\Delta_m = \{p \in \mathbb{R}^m : p_i \geq 0, \sum_i p_i = 1\}$ is the $m$-dimensional probability simplex. Then consider the function $h_{\vec{g},\eta} : \mathcal{X} \to \Delta(\mathcal{Y})$ with assignment probabilities given by $\Pr(h_{\vec{g},\eta}(x) = y) = \sum_{i=1}^{m} \mathbb{I}\{g_i(x) = y\}\eta_i$. Intuitively, for a given individual $x$, $h_{\vec{g},\eta}$ chooses one of the $g_i$ randomly with probability $\eta_i$, and outputs $g_i(x)$. Let

$$\mathcal{H}(\mathcal{G}, m) = \{h_{\vec{g},\eta} : \mathcal{X} \to \Delta(\mathcal{Y}) \,:\, \vec{g} \in \mathcal{G}^m, \eta \in \Delta_m\}$$

be the family of classifiers that can be written this way. Our main technical result shows that envy-freeness generalizes for this class.

**Theorem 3.** *Suppose $\mathcal{G}$ is a family of deterministic classifiers of Natarajan dimension $d$, and let $\mathcal{H} = \mathcal{H}(\mathcal{G}, m)$ for $m \in \mathbb{N}$. For any distribution $P$ over $\mathcal{X}$, $\gamma > 0$, and $\delta > 0$, if $S = \{(x_i, x_i')\}_{i=1}^{n}$ is an i.i.d. sample of pairs drawn from $P$ of size*

$$n \geq O\left(\frac{1}{\gamma^2}\left(dm^2 \log \frac{dm|\mathcal{Y}| \log(m|\mathcal{Y}|/\gamma)}{\gamma} + \log \frac{1}{\gamma}\right)\right),$$

*then with probability at least $1 - \delta$, every classifier $h \in \mathcal{H}$ that is $(\alpha, \beta)$-pairwise-EF on $S$ is also $(\alpha + 7\gamma, \beta + 4\gamma)$-EF on $P$.*

The proof of Theorem 3 is relegated to Appendix C. In a nutshell, it consists of two steps. First, we show that envy-freeness generalizes for finite classes. Second, we show that $\mathcal{H}(\mathcal{G}, m)$ can be approximated by a finite subset.

We remark that the theorem is only effective insofar as families of classifiers of low Natarajan dimension are useful. Fortunately, several prominent families indeed have low Natarajan dimension [5], including one vs. all, multiclass SVM, tree-based classifiers, and error correcting output codes.

# 5 Implementation and Empirical Validation

So far we have not directly addressed the problem of *computing* the loss-minimizing envy-free classifier from a given family on a given sample of individuals. We now turn to this problem. Our goal is not to provide an end-all solution, but rather to provide evidence that computation will not be a long-term obstacle to implementing our approach.

In more detail, our computational problem is to find the loss-minimizing classifier $h$ from a given family of randomized classifiers $\mathcal{H}$ that is envy free on a given a sample of individuals $S = \{x_1, \ldots, x_n\}$. For this classifier $h$ to generalize to the distribution $P$, Theorem 3 suggests that the family $\mathcal{H}$ to use is of the form $\mathcal{H}(\mathcal{G}, m)$, where $\mathcal{G}$ is a family of deterministic classifiers of low Natarajan dimension.

In this section, we let $\mathcal{G}$ be the family of *linear one-vs-all classifiers*. In particular, denoting $\mathcal{X} \subset \mathbb{R}^q$, each $g \in \mathcal{G}$ is parameterized by $\vec{w} = (w_1, w_2, \ldots, w_{|\mathcal{Y}|}) \in \mathbb{R}^{|\mathcal{Y}| \times q}$, where $g(x) = \text{argmax}_{y \in \mathcal{Y}} (w_y^\top x)$. This class $\mathcal{G}$ has a Natarajan dimension of at most $q|\mathcal{Y}|$. The optimization

problem to solve in this case is

$$\min_{\vec{g}\in\mathcal{G}^m,\eta\in\Delta_m}\quad \sum_{i=1}^{n}\sum_{k=1}^{m}\eta_k L(x_i,g_k(x_i))$$

$$\text{s.t.}\quad \sum_{k=1}^{m}\eta_k u(x_i,g_k(x_i)) \geq \sum_{k=1}^{m}\eta_k u(x_i,g_k(x_j)) \quad \forall(i,j)\in[n]^2. \tag{1}$$

### 5.1 Algorithm

Observe that optimization problem (1) is highly non-convex and non-differentiable as formulated, because of the argmax computed in each of the $g_k(x_i)$. Another challenge is the combinatorial nature of the problem, as we need to find $m$ functions from $\mathcal{G}$ along with their mixing weights. In designing an algorithm, therefore, we employ several tricks of the trade to achieve tractability.

**Learning the mixture components.** We first assume predefined mixing weights $\tilde{\eta}$, and *iteratively* learn mixture components based on them. Specifically, let $g_1, g_2, \ldots g_{k-1}$ denote the classifiers learned so far. To compute the next component $g_k$, we solve the optimization problem (1) with these components already in place (and assuming no future ones). This induces the following optimization problem.

$$\min_{g_k\in\mathcal{G}}\quad \sum_{i=1}^{n}L(x_i,g_k(x_i))$$

$$\text{s.t.}\quad USF_{ii}^{(k-1)} + \tilde{\eta}_k u(x_i,g_k(x_i)) \geq USF_{ij}^{(k-1)} + \tilde{\eta}_k u(x_i,g_k(x_j)) \quad \forall(i,j)\in[n]^2, \tag{2}$$

where $USF_{ij}^{(k-1)}$ denotes the expected utility $i$ has for $j$'s assignments so far, i.e., $USF_{ij}^{(k-1)} = \sum_{c=1}^{k-1}\tilde{\eta}_c u(x_i,g_c(x_j))$.

Solving the optimization problem (2) is still non-trivial because it remains non-convex and non-differentiable. To resolve this, we first soften the constraints[3]. Writing out the optimization problem in the form equivalent to introducing slack variables, we obtain

$$\min_{g_k\in\mathcal{G}}\quad \sum_{i=1}^{n}L(x_i,g_k(x_i))$$

$$+\lambda\sum_{i\neq j}\max\left(USF_{ij}^{(k-1)} + \tilde{\eta}_k u(x_i,g_k(x_j)) - USF_{ii}^{(k-1)} - \tilde{\eta}_k u(x_i,g_k(x_i)), 0\right), \tag{3}$$

where $\lambda$ is a parameter that defines the trade-off between loss and envy-freeness. This optimization problem is still highly non-convex as $g_k(x_i) = \text{argmax}_{y\in\mathcal{Y}}w_y^\top x_i$, where $\vec{w}$ denotes the parameters of $g_k$. To solve this, we perform a convex relaxation on several components of the objective using the fact that $w_{g_k(x_i)}^\top x_i \geq w_{y'}^\top x_i$ for any $y' \in \mathcal{Y}$. Specifically, we have

$$L(x_i,g_k(x_i)) \leq \max_{y\in\mathcal{Y}}\left\{L(x_i,y) + w_y^\top x_i - w_{y_i}^\top x_i\right\},$$

$$-u(x_i,g_k(x_i)) \leq \max_{y\in\mathcal{Y}}\left\{-u(x_i,y) + w_y^\top x_i - w_{b_i}^\top x_i\right\}, \text{ and}$$

$$u(x_i,g_k(x_j)) \leq \max_{y\in\mathcal{Y}}\left\{u(x_i,y) + w_y^\top x_j - w_{s_i}^\top x_j\right\},$$

where $y_i = \text{argmin}_{y\in\mathcal{Y}}L(x_i,y)$, $s_i = \text{argmin}_{y\in\mathcal{Y}}u(x_i,y)$ and $b_i = \text{argmax}_{y\in\mathcal{Y}}u(x_i,y)$. While we provided the key steps here, complete details and the rationale behind these choices are given in Appendix D. On a very high-level, these are inspired by multi-class SVMs. Finally, plugging these relaxations into (3), we obtain the following convex optimization problem to compute each mixture component.

$$\min_{\vec{w}\in\mathbb{R}^{|\mathcal{Y}|\times q}}\quad \sum_{i=1}^{n}\max_{y\in\mathcal{Y}}\left\{L(x_i,y) + w_y^\top x_i - w_{y_i}^\top x_i\right\} + \lambda\sum_{i\neq j}\max\left(USF_{ij}^{(k-1)}\right. \tag{4}$$

$$\left. +\tilde{\eta}_k\max_{y\in\mathcal{Y}}\left\{u(x_i,y) + w_y^\top x_j - w_{s_i}^\top x_j\right\} - USF_{ii}^{(k-1)} + \tilde{\eta}_k\max_{y\in\mathcal{Y}}\left\{-u(x_i,y) + w_y^\top x_i - w_{b_i}^\top x_i\right\}, 0\right).$$

**Learning the mixing weights.** Once the mixture components $\vec{g}$ are learned (with respect to the predefined mixing weights $\tilde{\eta}$), we perform an additional round of optimization to learn the optimal weights $\eta$ for them. This can be done via the following linear program

$$\min_{\eta \in \Delta_m, \xi \in \mathbb{R}^{n \times n}_{\geq 0}} \quad \sum_{i=1}^{n} \sum_{k=1}^{m} \eta_k L(x_i, g_k(x_i)) + \lambda \sum_{i \neq j} \xi_{ij}$$

$$\text{s.t.} \quad \sum_{k=1}^{m} \eta_k u(x_i, g_k(x_i)) \geq \sum_{k=1}^{m} \eta_k u(x_i, g_k(x_j)) - \xi_{ij} \quad \forall (i,j). \tag{5}$$

## 5.2 Methodology

To validate our approach, we have implemented our algorithm. However, we cannot rely on standard datasets, as we need access to both the features and the utility functions of individuals. Hence, we rely on synthetic data. All our code is included as supplementary material. Our experiments are carried out on a desktop machine with 16GB memory and an Intel Xeon(R) CPU E5-1603 v3 @ 2.80GHz×4 processor. To solve convex optimization problems, we use CVXPY [7, 1].

In our experiments, we cannot compute the optimal solution to the original optimization problem (1), and there are no existing methods we can use as benchmarks. Hence, we generate the dataset such that we know the optimal solution upfront.

Specifically, to generate the whole dataset (both training and test), we first generate random classifiers $\vec{g}^\star \in \mathcal{G}^m$ by sampling their parameters $\vec{w}_1, \ldots \vec{w}_m \sim \mathcal{N}(0,1)^{|\mathcal{Y}| \times q}$, and generate $\eta^\star \in \Delta_m$ by drawing uniformly random weights in $[0,1]$ and normalizing. We use $h_{\vec{g}^\star, \eta^\star}$ as the optimal solution of the dataset we generate. For each individual, we sample each feature value independently and u.a.r. in $[0,1]$. For each individual $x$ and outcome $y$, we set $L(x,y) = 0$ if $y \in \{g_k^\star(x) : k \in [m]\}$ and otherwise we sample $L(x,y)$ u.a.r. in $[0,1]$. For the utility function $u$, we need to generate it such that the randomized classifier $h_{\vec{g}^\star, \eta^\star}$ is envy free on the dataset. For this, we set up a linear program and compute each of the values $u(x,y)$. Hence, $h_{\vec{g}^\star, \eta^\star}$ is envy free and has zero loss, so it is obviously the optimal solution. The dataset is split into 75% training data (to measure the accuracy of our solution to the optimization problem) and 25% test data (to evaluate generalizability).

For our experiments, we use the following parameters: $|\mathcal{Y}| = 10$, $q = 10$, $m = 5$, and $\lambda = 10.0$. We set the predefined weights to be $\tilde{\eta} = \left[\frac{1}{2}, \frac{1}{4}, \ldots, \frac{1}{2^{m-1}}, \frac{1}{2^{m-1}}\right]$.[4] In our experiments we vary the number of individuals, and each result is averaged over 25 runs. On each run, we generate a new ground-truth classifier $h_{\vec{g}^*, \eta^*}$, as well as new individuals, losses, and utilities.

## 5.3 Results

Figure 1 shows the time taken to compute the mixture components $\vec{g}$ and the optimal weights $\eta$, as the number of individuals in the training data increases. As we will see shortly, even though the $\eta$ computation takes a very small fraction of the time, it can lead to non-negligible gains in terms of loss and envy.

Figure 2 shows the average loss attained on the training and test data by the algorithm immediately after computing the mixture components, and after the round of $\eta$ optimization. It also shows the average loss attained (on both the training and test data) by a random allocation, which serves as a naïve benchmark for calibration purposes. Recall that the optimal assignment $h_{\vec{g}^\star, \eta^\star}$ has loss 0. For both the training and testing individuals, optimizing $\eta$ improves the loss of the learned classifer. Moreover, our algorithms achieve low training errors for all dataset sizes, and as the dataset grows the testing error converges to the training error.

Figure 3 shows the average envy among pairs in the training data and test data, where, for each pair, negative envy is replaced with 0, to avoid obfuscating positive envy. The graph also depicts the average envy attained (on both the training and test data) by a random allocation. As for the losses, optimizing $\eta$ results in lower average envy, and as the training set grows we see the generalization gap decrease.

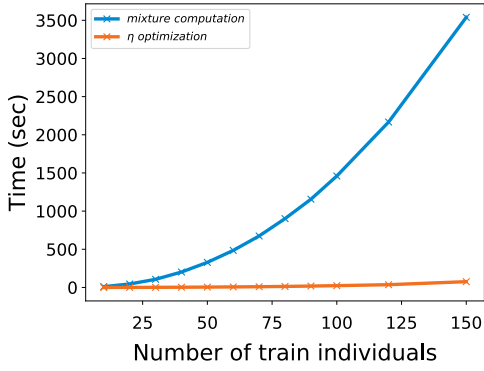

Figure 1: The algorithm's running time.

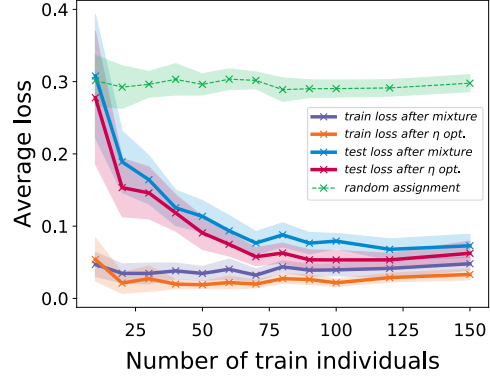

Figure 2: Training and test loss. Shaded error bands depict $95\%$ confidence intervals.

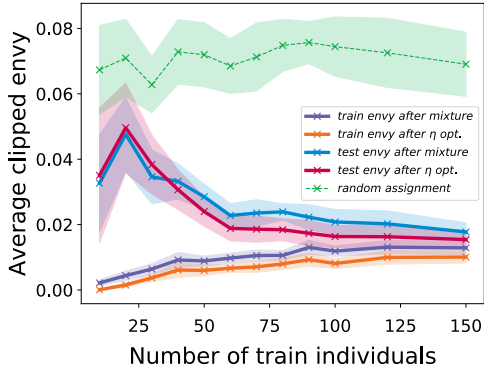

Figure 3: Training and test envy, as a function of the number of individuals. Shaded error bands depict $95\%$ confidence intervals.

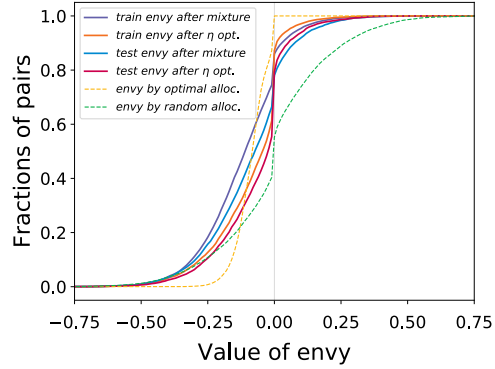

Figure 4: CDF of training and test envy for 100 training individuals

In Figure 4 we zoom in on the case of 100 training individuals, and observe the empirical CDF of envy values. Interestingly, the optimal randomized classifier $h_{\vec{g}^\star, \eta^\star}$ shows lower negative envy values compared to other algorithms, but as expected has no positive envy pairs. Looking at the positive envy values, we can again see very encouraging results. In particular, for at least a $0.946$ fraction of the pairs in the train data, we obtain envy of at most $0.05$, and this generalizes to the test data, where for at least a $0.939$ fraction of the pairs, we obtain envy of at most $0.1$.

In summary, these results indicate that the algorithm described in Section 5.1 solves the optimization problem (1) for linear one-vs-all classifiers almost optimally, and that its output generalizes well even when the training set is small.

# 6   Conclusion

In this paper we propose EF as a suitable fairness notion for learning tasks with many outcomes over which individuals have heterogeneous preferences. We provide generalization guarantees for a rich family of classifiers, showing that if we find a classifier that is envy-free on a sample of individuals, it will remain envy-free when we apply it to new individuals from the same distribution. This result circumvents an exponential lower bound on the sample complexity suffered by any two-stage learning algorithm that first finds an EF assignment for the sample and then extends it to the entire space. Finally, we empirically demonstrate that finding low-envy and low-loss classifiers is computationally tractable. These results show that envy-freeness is a practical notion of fairness for machine learning systems.

## Acknowledgments

This work was partially supported by the National Science Foundation under grants IIS-1350598, IIS-1714140, IIS-1618714, IIS-1901403, CCF-1525932, CCF-1733556, CCF-1535967, CCF-1910321; by the Office of Naval Research under grants N00014-16-1-3075 and N00014-17-1-2428; and by a J.P. Morgan AI Research Award, an Amazon Research Award, a Microsoft Research Faculty Fellowship, a Bloomberg Data Science research grant, a Guggenheim Fellowship, and a grant from the Block Center for Technology and Society.

## Footnotes

[1]Certain papers take a somewhat different view [17].

[2]On a philosophical level, the fair division literature deals exclusively with individual notions of fairness. In fact, even in group-based extensions of envy-freeness [19] the allocation is shared by groups, but individuals must not be envious. We subscribe to the view that group-oriented notions (such as statistical parity) are objectionable, because the outcome can be patently unfair to individuals.

[3]This may lead to solutions that are not exactly EF on the sample. Nonetheless, Theorem 3 still guarantees that there should not be much additional envy on the testing data.

[4]The reason for using an exponential decay is so that the subsequent classifiers learned are different from the previous ones. Using smaller weights might cause consecutive classifiers to be identical, thereby 'wasting' some of the components.

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
