[Supplementary Material · 742_appendix.pdf]

# Appendix: Envy-Free Classification

## A    Natarajan Dimension Primer

We briefly present the Natarajan dimension. For more details, we refer the reader to [27].

We say that a family $\mathcal{G}$ *multi-class shatters* a set of points $x_1, \ldots, x_n$ if there exist labels $y_1, \ldots y_n$ and $y_1', \ldots, y_n'$ such that for every $i \in [n]$ we have $y_i \neq y_i'$, and for any subset $C \subset [n]$ there exists $g \in \mathcal{G}$ such that $g(x_i) = y_i$ if $i \in C$ and $g(x_i) = y_i'$ otherwise. The Natarajan dimension of a family $\mathcal{G}$ is the cardinality of the largest set of points that can be multi-class shattered by $\mathcal{G}$.

For example, suppose we have a feature map $\Psi : \mathcal{X} \times \mathcal{Y} \to \mathbb{R}^q$ that maps each individual-outcome pair to a $q$-dimensional feature vector, and consider the family of functions that can be written as $g(x) = \arg\max_{y \in \mathcal{Y}} w^\top \Psi(x, y)$ for weight vectors $w \in \mathbb{R}^q$. This family has Natarajan dimension at most $q$.

For a set $S \subset \mathcal{X}$ of points, we let $\mathcal{G}\big|_S$ denote the restriction of $\mathcal{G}$ to $S$, which is any subset of $\mathcal{G}$ of minimal size such that for every $g \in \mathcal{G}$ there exists $g' \in \mathcal{G}\big|_S$ such that $g(x) = g'(x)$ for all $x \in S$. The size of $\mathcal{G}\big|_S$ is the number of different labelings of the sample $S$ achievable by functions in $\mathcal{G}$. The following Lemma is the analogue of Sauer's lemma for binary classification.

**Lemma 1** (Natarajan). *For a family $\mathcal{G}$ of Natarajan dimension $d$ and any subset $S \subset \mathcal{X}$, we have* $\big|\mathcal{G}\big|_S\big| \leq |S|^d |\mathcal{Y}|^{2d}$.

Classes of low Natarajan dimension also enjoy the following uniform convergence guarantee.

**Lemma 2.** *Let $\mathcal{G}$ have Natarajan dimension $d$ and fix a loss function $\ell : \mathcal{G} \times \mathcal{X} \to [0, 1]$. For any distribution $P$ over $\mathcal{X}$, if $S$ is an i.i.d. sample drawn from $P$ of size $O(\frac{1}{\epsilon^2}(d \log |\mathcal{Y}| + \log \frac{1}{\delta}))$, then with probability at least $1 - \delta$ we have $\sup_{g \in \mathcal{G}} \big|\mathbb{E}_{x \sim P}[\ell(g, x)] - \frac{1}{n} \sum_{x \in S} \ell(g, x)\big| \leq \epsilon$.*

## B    Appendix for Section 3

**Theorem 1.** *Let $d$ be a metric on $\mathcal{X}$, $P$ be a distribution on $\mathcal{X}$, and $u$ be an $L$-Lipschitz utility function. Let $S$ be a set of individuals such that there exists $\hat{\mathcal{X}} \subset \mathcal{X}$ with $P(\hat{\mathcal{X}}) \geq 1 - \alpha$ and $\sup_{x \in \hat{\mathcal{X}}} d(x, \mathrm{NN}_S(x)) \leq \beta/(2L)$. Then for any classifier $h : S \to \Delta(\mathcal{Y})$ that is EF on $S$, the extension $\overline{h} : \mathcal{X} \to \Delta(\mathcal{Y})$ given by $\overline{h}(x) = h(\mathrm{NN}_S(x))$ is $(\alpha, \beta)$-EF on $P$.*

*Proof.* Let $h : S \to \Delta(\mathcal{Y})$ be any EF classifier on $S$ and $\overline{h} : \mathcal{X} \to \Delta(\mathcal{Y})$ be the nearest neighbor extension. Sample $x$ and $x'$ from $P$. Then, $x$ belongs to the subset $\hat{\mathcal{X}}$ with probability at least $1 - \alpha$. When this occurs, $x$ has a neighbor within distance $\beta/(2L)$ in the sample. Using the Lipschitz continuity of $u$, we have $|u(x, \overline{h}(x)) - u(\mathrm{NN}_S(x), h(\mathrm{NN}_S(x)))| \leq \beta/2$. Similarly, $|u(x, \overline{h}(x')) - u(\mathrm{NN}_S(x), h(\mathrm{NN}_S(x')))| \leq \beta/2$. Finally, since $\mathrm{NN}_S(x)$ does not envy $\mathrm{NN}_S(x')$ under $h$, it follows that $x$ does not envy $x'$ by more than $\beta$ under $\overline{h}$. □

**Lemma 3.** *Suppose $\mathcal{X} \subset \mathbb{R}^q$, $d(x, x') = \|x - x'\|_2$, and let $D = \sup_{x, x' \in \mathcal{X}} d(x, x')$ be the diameter of $\mathcal{X}$. For any distribution $P$ over $\mathcal{X}$, $\beta > 0$, $\alpha > 0$, and $\delta > 0$ there exists $\hat{\mathcal{X}} \subset \mathcal{X}$ such that $P(\hat{\mathcal{X}}) \geq 1 - \alpha$ and, if $S$ is an i.i.d. sample drawn from $P$ of size $|S| = O(\frac{1}{\alpha}(\frac{LD\sqrt{q}}{\beta})^q (d \log \frac{LD\sqrt{q}}{\beta} + \log \frac{1}{\delta}))$, then with probability at least $1 - \delta$, $\sup_{x \in \hat{\mathcal{X}}} d(x, \mathrm{NN}_S(x)) \leq \beta/(2L)$.*

*Proof.* Let $C$ be the smallest cube containing $\mathcal{X}$. Since the diameter of $\mathcal{X}$ is $D$, the side-length of $C$ is at most $D$. Let $s = \beta/(2L\sqrt{q})$ be the side-length such that a cube with side-length $s$ has diameter $\beta/(2L)$. It takes at most $m = \lceil D/s \rceil^q$ cubes of side-length $s$ to cover $C$. Let $C_1, \ldots, C_m$ be such a covering, where each $C_i$ has side-length $s$.

Let $C_i$ be any cube in the cover for which $P(C_i) > \alpha/m$. The probability that a sample of size $n$ drawn from $P$ does not contain a sample in $C_i$ is at most $(1 - \alpha/m)^n \leq e^{-n\alpha/m}$. Let

$I = \{i \in [m] : P(C_i) \geq \alpha/m\}$. By the union bound, the probability that there exists $i \in I$ such that $C_i$ does not contain a sample is at most $me^{-n\alpha/m}$. Setting

$$n = \frac{m}{\alpha} \ln \frac{m}{\delta}$$

$$= O\left(\frac{1}{\alpha}\left(\frac{LD\sqrt{q}}{\beta}\right)^q \left(q \log \frac{LD\sqrt{q}}{\beta} + \log \frac{1}{\delta}\right)\right)$$

results in this upper bound being $\delta$. For the remainder of the proof, assume this high probability event occurs.

Now let $\hat{\mathcal{X}} = \bigcup_{i \in I} C_i$. For each $j \notin I$, we know that $P(C_j) < \alpha/m$. Since there at most $m$ such cubes, their total probability mass is at most $\alpha$. It follows that $P(\hat{\mathcal{X}}) \geq 1 - \alpha$. Moreover, every point $x \in \hat{\mathcal{X}}$ belongs to one of the cubes $C_i$ with $i \in I$, which also contains a sample point. Since the diameter of the cubes in our cover is $\beta/(2L)$, it follows that $\text{dist}(x, \text{NN}_S(x)) \leq \beta/(2L)$ for every $x \in \hat{\mathcal{X}}$, as required. □

**Theorem 2.** *There exists a space of individuals $\mathcal{X} \subset \mathbb{R}^q$, and a distribution $P$ over $\mathcal{X}$ such that, for every randomized algorithm $\mathcal{A}$ that extends classifiers on a sample to $\mathcal{X}$, there exists an $L$-Lipschitz utility function $u$ such that, when a sample of individuals $S$ of size $n = 4^q/2$ is drawn from $P$ without replacement, there exists an EF classifier on $S$ for which, with probability at least $1 - 2\exp(-4^q/100) - \exp(-4^q/200)$ jointly over the randomness of $\mathcal{A}$ and $S$, its extension by $\mathcal{A}$ is not $(\alpha, \beta)$-EF with respect to $P$ for any $\alpha < 1/25$ and $\beta < L/8$.*

*Proof.* Let the space of individuals be $\mathcal{X} = [0,1]^q$ and the outcomes be $\mathcal{Y} = \{0,1\}$. We partition the space $\mathcal{X}$ into cubes of side length $s = 1/4$. So, the total number of cubes is $m = (1/s)^q = 4^q$. Let these cubes be denoted by $c_1, c_2, \ldots c_m$, and let their centers be denoted by $\mu_1, \mu_2, \ldots \mu_m$. Next, let $P$ be the uniform distribution over the centers $\mu_1, \mu_2, \ldots \mu_m$. For brevity, whenever we say "utility function" in the rest of the proof, we mean "$L$-Lipschitz utility function."

To prove the theorem, we use Yao's minimax principle [33]. Specifically, consider the following two-player zero sum game. Player 1 chooses a deterministic algorithm $\mathcal{D}$ that extends classifiers on a sample to $\mathcal{X}$, and player 2 chooses a utility function $u$ on $\mathcal{X}$. For any subset $S \subset \mathcal{X}$, define the classifier $h_{u,S} : S \to \mathcal{Y}$ by assigning each individual in $S$ to his favorite outcome with respect to the utility function $u$, i.e. $h_{u,S}(x) = \arg\max_{y \in \mathcal{Y}} u(x,y)$ for each $x \in S$, breaking ties lexicographically. Define the cost of playing algorithm $\mathcal{D}$ against utility function $u$ as the probability over the sample $S$ (of size $m/2$ drawn from $P$ without replacement) that the extension of $h_{u,S}$ by $\mathcal{D}$ is not $(\alpha, \beta)$-EF with respect to $P$ for any $\alpha < 1/25$ and $\beta < L/8$. Yao's minimax principle implies that for any randomized algorithm $\mathcal{A}$, its expected cost with respect to the worst-case utility function $u$ is at least as high as the expected cost of any distribution over utility functions that is played against the best deterministic algorithm $\mathcal{D}$ (which is tailored for that distribution). Therefore, we establish the desired lower bound by choosing a specific distribution over utility functions, and showing that the best deterministic algorithm against it has an expected cost of at least $1 - 2\exp(-m/100) - \exp(-m/200)$.

To define this distribution over utility functions, we first sample outcomes $y_1, y_2, \ldots, y_m$ i.i.d. from Bernoulli(1/2). Then, we associate each cube center $\mu_i$ with the outcome $y_i$, and refer to this outcome as the *favorite* of $\mu_i$. For brevity, let $\neg y$ denote the outcome other than $y$, i.e. $\neg y = (1 - y)$. For any $x \in \mathcal{X}$, we define the utility function as follows. Letting $c_j$ be the cube that $x$ belongs to,

$$u(x, y_j) = L\left[\frac{s}{2} - \|x - \mu_j\|_\infty\right]; \quad u(x, \neg y_j) = 0. \tag{6}$$

See Figure 5 for an illustration.

We claim that the utility function of Equation (6) is indeed $L$-Lipschitz with respect to any $L_p$ norm. This is because for any cube $c_i$, and for any $x, x' \in c_i$, we have

$$|u(x, y_i) - u(x', y_i)| = L\left|\|x - \mu_i\|_\infty - \|x' - \mu_i\|_\infty\right|$$

$$\leq L\|x - x'\|_\infty \leq L\|x - x'\|_p.$$

Moreover, for the other outcome, we have $u(x, \neg y_i) = u(x', \neg y_i) = 0$. It follows that $u$ is $L$-Lipschitz within every cube. At the boundary of the cubes, the utility for any outcome is 0, and hence

Figure 5: Illustration of $\mathcal{X}$ and an example utility function $u$ for $d = 2$. Red shows preference for 1, blue shows preference for 0, and darker shades correspond to more intense preference. (The gradients are rectangular to match the $L_\infty$ norm, so, strangely enough, the misleading X pattern is an optical illusion.)

$u$ is also continuous throughout $\mathcal{X}$. Because it is piecewise Lipschitz and continuous, $u$ must be $L$-Lipschitz throughout $\mathcal{X}$, with respect to any $L_p$ norm.

Next, let $\mathcal{D}$ be an arbitrary deterministic algorithm that extends classifiers on a sample to $\mathcal{X}$. We draw the sample $S$ of size $m/2$ from $P$ without replacement. Consider the distribution over favorites of individuals in $S$. Each individual in $S$ has a favorite that is sampled independently from Bernoulli$(1/2)$. Hence, by Hoeffding's inequality, the fraction of individuals in $S$ with a favorite of 0 is between $\frac{1}{2} - \epsilon$ and $\frac{1}{2} + \epsilon$ with probability at least $1 - 2\exp(-m\epsilon^2)$. The same holds simultaneously for the fraction of individuals with favorite 1.

Given the sample $S$ and the utility function $u$ on the sample (defined by the instantiation of their favorites), consider the classifier $h_{u,S}$, which maps each individual $\mu_i$ in the sample $S$ to his favorite $y_i$. This classifier is clearly EF on the sample. Consider the extension $h_{u,S}^{\mathcal{D}}$ of $h_{u,S}$ to the whole of $\mathcal{X}$ as defined by algorithm $\mathcal{D}$. Define two sets $Z_0$ and $Z_1$ by letting $Z_y = \{\mu_j \notin S \mid h_{u,S}^{\mathcal{D}}(\mu_j) = y\}$, and let $y_*$ denote an outcome that is assigned to at least half of the out-of-sample centers, i.e., an outcome for which $|Z_{y_*}| \geq |Z_{\neg y_*}|$. Furthermore, let $\theta$ denote the fraction of out-of-sample centers assigned to $y_*$. Note that, since $|S| = m/2$, the number of out-of-sample centers is also exactly $m/2$. This gives us $|Z_{y_*}| = \theta\frac{m}{2}$, where $\theta \geq \frac{1}{2}$.

Consider the distribution of favorites in $Z_{y_*}$ (these are independent from the ones in the sample since $Z_{y_*}$ is disjoint from $S$). Each individual in this set has a favorite sampled independently from Bernoulli$(1/2)$. Hence, by Hoeffding's inequality, the fraction of individuals in $Z_{y_*}$ whose favorite is $\neg y_*$ is at least $\frac{1}{2} - \epsilon$ with probability at least $1 - \exp(-\frac{m}{2}\epsilon^2)$. We conclude that with a probability at least $1 - 2\exp(-m\epsilon^2) - \exp(-\frac{m}{2}\epsilon^2)$, the sample $S$ and favorites (which define the utility function $u$) are such that: (i) the fraction of individuals in $S$ whose favorite is $y \in \{0, 1\}$ is between $\frac{1}{2} - \epsilon$ and $\frac{1}{2} + \epsilon$, and (ii) the fraction of individuals in $Z_{y_*}$ whose favorite is $\neg y_*$ is at least $\frac{1}{2} - \epsilon$.

We now show that for such a sample $S$ and utility function $u$, $h_{u,S}^{\mathcal{D}}$ cannot be $(\alpha, \beta)$-EF with respect to $P$ for any $\alpha < 1/25$ and $\beta < L/8$. To this end, sample $x$ and $x'$ from $P$. One scenario where $x$ envies $x'$ occurs when (i) the favorite of $x$ is $\neg y_*$, (ii) $x$ is assigned to $y_*$, and (iii) $x'$ is assigned to $\neg y_*$. Conditions (i) and (ii) are satisfied when $x$ is in $Z_{y_*}$ and his favorite is $\neg y_*$. We know that at least a $\frac{1}{2} - \epsilon$ fraction of the individuals in $Z_{y_*}$ have the favorite $\neg y_*$. Hence, the probability that conditions (i) and (ii) are satisfied by $x$ is at least $(\frac{1}{2} - \epsilon)|Z_{y_*}|\frac{1}{m} = (\frac{1}{2} - \epsilon)\frac{\theta}{2}$. Condition (iii) is satisfied when $x'$ is in $S$ and has favorite $\neg y_*$ (and hence assigned $\neg y_*$), or, if $x'$ is in $Z_{\neg y_*}$. We know that at least a $(\frac{1}{2} - \epsilon)$ fraction of the individuals in $S$ have the favorite $\neg y_*$. Moreover, the size of $Z_{\neg y_*}$ is $(1 - \theta)\frac{m}{2}$. So, the probability that condition (iii) is satisfied by $x'$ is at least

$$\frac{\left(\frac{1}{2} - \epsilon\right)|S| + |Z_{\neg y_*}|}{m} = \frac{1}{2}\left(\frac{1}{2} - \epsilon\right) + \frac{1}{2}(1 - \theta).$$

Since $x$ and $x'$ are sampled independently, the probability that all three conditions are satisfied is at least

$$\left(\frac{1}{2} - \epsilon\right) \frac{\theta}{2} \cdot \left[\frac{1}{2}\left(\frac{1}{2} - \epsilon\right) + \frac{1}{2}(1 - \theta)\right].$$

This expression is a quadratic function in $\theta$, that attains its minimum at $\theta = 1$ irrespective of the value of $\epsilon$. Hence, irrespective of $\mathcal{D}$, this probability is at least $\left[\frac{1}{2}\left(\frac{1}{2} - \epsilon\right)\right]^2$. For concreteness, let us choose $\epsilon$ to be $1/10$ (although it can be set to be much smaller). On doing so, we have that the three conditions are satisfied with probability at least $1/25$. And when these conditions are satisfied, we have $u(x, h^{\mathcal{D}}_{u,S}(x)) = 0$ and $u(x, h^{\mathcal{D}}_{u,S}(x')) = Ls/2$, i.e., $x$ envies $x'$ by $Ls/2 = L/8$. This shows that, when $x$ and $x'$ are sampled from $P$, with probability at least $1/25$, $x$ envies $x'$ by $L/8$. We conclude that with probability at least $1 - 2\exp(-m/100) - \exp(-m/200)$ jointly over the selection of the utility function $u$ and the sample $S$, the extension of $h_{u,S}$ by $\mathcal{D}$ is not $(\alpha, \beta)$-EF with respect to $P$ for any $\alpha < 1/25$ and $\beta < L/8$.

To convert the joint probability into expected cost in the game, note that for two discrete, independent random variables $X$ and $Y$, and for a Boolean function $\mathcal{E}(X, Y)$, it holds that

$$\Pr_{X,Y}(\mathcal{E}(X, Y) = 1) = \mathbb{E}_X\left[\Pr_Y(\mathcal{E}(X, Y) = 1)\right]. \tag{7}$$

Given sample $S$ and utility function $u$, let $\mathcal{E}(u, S)$ be the Boolean function that equals 1 if and only if the extension of $h_{u,S}$ by $\mathcal{D}$ is not $(\alpha, \beta)$-EF with respect to $P$ for any $\alpha < 1/25$ and $\beta < L/8$. From Equation (7), $\Pr_{u,S}(\mathcal{E}(u, S) = 1)$ is equal to $\mathbb{E}_u\left[\Pr_S(\mathcal{E}(u, S) = 1)\right]$. The latter term is exactly the expected value of the cost, where the expectation is taken over the randomness of $u$. It follows that the expected cost of (any) $\mathcal{D}$ with respect to the chosen distribution over utilities is at least $1 - 2\exp(-m/100) - \exp(-m/200)$. $\qquad\square$

## C   Appendix for Section 4

This section is devoted to proving our main result:

**Theorem 3.** *Suppose $\mathcal{G}$ is a family of deterministic classifiers of Natarajan dimension $d$, and let $\mathcal{H} = \mathcal{H}(\mathcal{G}, m)$ for $m \in \mathbb{N}$. For any distribution $P$ over $\mathcal{X}$, $\gamma > 0$, and $\delta > 0$, if $S = \{(x_i, x'_i)\}^n_{i=1}$ is an i.i.d. sample of pairs drawn from $P$ of size*

$$n \geq O\left(\frac{1}{\gamma^2}\left(dm^2 \log \frac{dm|\mathcal{Y}| \log(m|\mathcal{Y}|/\gamma)}{\gamma} + \log \frac{1}{\gamma}\right)\right),$$

*then with probability at least $1 - \delta$, every classifier $h \in \mathcal{H}$ that is $(\alpha, \beta)$-pairwise-EF on $S$ is also $(\alpha + 7\gamma, \beta + 4\gamma)$-EF on $P$.*

We start with an observation that will be required later.

**Lemma 4.** *Let $\mathcal{G} = \{g : \mathcal{X} \to \mathcal{Y}\}$ have Natarajan dimension $d$. For $g_1, g_2 \in \mathcal{G}$, let $(g_1, g_2) : \mathcal{X} \to \mathcal{Y}^2$ denote the function given by $(g_1, g_2)(x) = (g_1(x), g_2(x))$ and let $\mathcal{G}^2 = \{(g_1, g_2) : g_1, g_2 \in \mathcal{G}\}$. Then the Natarajan dimension of $\mathcal{G}^2$ is at most $2d$.*

*Proof.* Let $D$ be the Natarajan dimension of $\mathcal{G}^2$. Then we know that there exists a collection of points $x_1, \ldots, x_D \in \mathcal{X}$ that is shattered by $\mathcal{G}^2$, which means there are two sequences $q_1, \ldots, q_n \in \mathcal{Y}^2$ and $q'_1, \ldots, q'_n \in \mathcal{Y}^2$ such that for all $i$ we have $q_i \neq q'_i$ and for any subset $C \subset [D]$ of indices, there exists $(g_1, g_2) \in \mathcal{G}^2$ such that $(g_1, g_2)(x_i) = q_i$ if $i \in C$ and $(g_1, g_2)(x_i) = q'_i$ otherwise.

Let $n_1 = \sum^D_{i=1} \mathbb{I}\{q_{i1} \neq q'_{i1}\}$ and $n_2 = \sum^D_{i=1} \mathbb{I}\{q_{i2} \neq q'_{i2}\}$ be the number of pairs on which the first and second labels of $q_i$ and $q'_i$ disagree, respectively. Since none of the $n$ pairs are equal, we know that $n_1 + n_2 \geq D$, which implies that at at least one of $n_1$ or $n_2$ must be $\geq D/2$. Assume without loss of generality that $n_1 \geq D/2$ and that $q_{i1} \neq q'_{i1}$ for $i = 1, \ldots, n_1$. Now consider any subset of indices $C \subset [n_1]$. We know there exists a pair of functions $(g_1, g_2) \in \mathcal{G}^2$ with $(g_1, g_2)(x_i)$ evaluating to $q_i$ if $i \in C$ and $q'_i$ if $i \notin C$. But then we have $g_1(x_i) = q_{i1}$ if $i \in C$ and $g_1(x_i) = q'_{i1}$ if $i \notin C$, and $q_{i1} \neq q'_{i1}$ for all $i \in [n_1]$. It follows that $\mathcal{G}$ shatters $x_1, \ldots, x_{n_1}$, which consists of at least $D/2$ points. Therefore, the Natarajan dimension of $\mathcal{G}^2$ is at most $2d$, as required. $\qquad\square$

We now turn two the theorem's two main steps, presented in the following two lemmas.

**Lemma 5.** *Let $\mathcal{H} \subset \{h : \mathcal{X} \to \Delta(\mathcal{Y})\}$ be a finite family of classifiers. For any $\gamma > 0$, $\delta > 0$, and $\beta \geq 0$ if $S = \{(x_i, x_i')\}_{i=1}^n$ is an i.i.d. sample of pairs from $P$ of size $n \geq \frac{1}{2\gamma^2} \ln \frac{|\mathcal{H}|}{\delta}$, then with probability at least $1 - \delta$, every $h \in \mathcal{H}$ that is $(\alpha, \beta)$-pairwise-EF on $S$ (for any $\alpha$) is also $(\alpha + \gamma, \beta)$-EF on $P$.*

*Proof.* Let $f(x, x', h) = \mathbb{I}\{u(x, h(x)) < u(x, h(x')) - \beta\}$ be the indicator that $x$ is envious of $x'$ by at least $\beta$ under classifier $h$. Then $f(x_i, x_i', h)$ is a Bernoulli random variable with success probability $\mathbb{E}_{x, x' \sim P}[f(x, x', h)]$. Applying Hoeffding's inequality to any fixed hypothesis $h \in \mathcal{H}$ guarantees that $\Pr_S(\mathbb{E}_{x, x' \sim P}[f(x, x', h)] \geq \frac{1}{n} \sum_{i=1}^n f(x_i, x_i', h) + \gamma) \leq \exp(-2n\gamma^2)$. Therefore, if $h$ is $(\alpha, \beta)$-EF on $S$, then it is also $(\alpha + \gamma, \beta)$-EF on $P$ with probability at least $1 - \exp(-2n\gamma^2)$. Applying the union bound over all $h \in \mathcal{H}$ and using the lower bound on $n$ completes the proof. $\qquad\square$

Next, we show that $\mathcal{H}(\mathcal{G}, m)$ can be covered by a finite subset. Since each classifier in $\mathcal{H}$ is determined by the choice of $m$ functions from $\mathcal{G}$ and mixing weights $\eta \in \Delta_m$, we will construct finite covers of $\mathcal{G}$ and $\Delta_m$. Our covers $\hat{\mathcal{G}}$ and $\hat{\Delta}_m$ will guarantee that for every $g \in \mathcal{G}$, there exists $\hat{g} \in \hat{\mathcal{G}}$ such that $\Pr_{x \sim P}(g(x) \neq \hat{g}(x)) \leq \gamma/m$. Similarly, for any mixing weights $\eta \in \Delta_m$, there exists $\hat{\eta} \in \Delta_m$ such that $\|\eta - \hat{\eta}\|_1 \leq \gamma$. If $h \in \mathcal{H}(\mathcal{G}, m)$ is the mixture of $g_1, \ldots, g_m$ with weights $\eta$, we let $\hat{h}$ be the mixture of $\hat{g}_1, \ldots, \hat{g}_m$ with weights $\hat{\eta}$. This approximation has two sources of error: first, for a random individual $x \sim P$, there is probability up to $\gamma$ that at least one $g_i(x)$ will disagree with $\hat{g}_i(x)$, in which case $h$ and $\hat{h}$ may assign completely different outcome distributions. Second, even in the high-probability event that $g_i(x) = \hat{g}_i(x)$ for all $i \in [m]$, the mixing weights are not identical, resulting in a small perturbation of the outcome distribution assigned to $x$.

**Lemma 6.** *Let $\mathcal{G}$ be a family of deterministic classifiers with Natarajan dimension $d$, and let $\mathcal{H} = \mathcal{H}(\mathcal{G}, m)$ for some $m \in \mathbb{N}$. For any $\gamma > 0$, there exists a subset $\hat{\mathcal{H}} \subset \mathcal{H}$ of size $O\left(\frac{(dm|\mathcal{Y}|^2 \log(m|\mathcal{Y}|/\gamma))^{dm}}{\gamma^{(d+1)m}}\right)$ such that for every $h \in \mathcal{H}$ there exists $\hat{h} \in \mathcal{H}$ satisfying:*

1. *$\Pr_{x \sim P}(\|h(x) - \hat{h}(x)\|_1 > \gamma) \leq \gamma$.*

2. *If $S$ is an i.i.d. sample of individuals of size $O(\frac{m^2}{\gamma^2}(d \log |\mathcal{Y}| + \log \frac{1}{\delta}))$ then w.p. $\geq 1 - \delta$, we have $\|h(x) - \hat{h}(x)\|_1 \leq \gamma$ for all but a $2\gamma$-fraction of $x \in S$.*

*Proof.* As described above, we begin by constructing finite covers of $\Delta_m$ and $\mathcal{G}$. First, let $\hat{\Delta}_m \subset \Delta_m$ be the set of distributions over $[m]$ where each coordinate is a multiple of $\gamma/m$. Then we have $|\hat{\Delta}_m| = O((\frac{m}{\gamma})^m)$ and for every $p \in \Delta_m$, there exists $q \in \hat{\Delta}_m$ such that $\|p - q\|_1 \leq \gamma$.

In order to find a small cover of $\mathcal{G}$, we use the fact that it has low Natarajan dimension. This implies that the number of effective functions in $\mathcal{G}$ when restricted to a sample $S'$ grows only polynomially in the size of $S'$. At the same time, if two functions in $\mathcal{G}$ agree on a large sample, they will also agree with high probability on the distribution.

Formally, let $S'$ be an i.i.d. sample drawn from $P$ of size $O(\frac{m^2}{\gamma^2} d \log |\mathcal{Y}|)$, and let $\hat{\mathcal{G}} = \mathcal{G}\big|_{S'}$ be any minimal subset of $\mathcal{G}$ that realizes all possible labelings of $S'$ by functions in $\mathcal{G}$. We now argue that with probability 0.99, for every $g \in \mathcal{G}$ there exists $\hat{g} \in \hat{\mathcal{G}}$ such that $\Pr_{x \sim P}(g(x) \neq \hat{g}(x)) \leq \gamma/m$. For any pair of functions $g, g' \in \mathcal{G}$, let $(g, g') : \mathcal{X} \to \mathcal{Y}^2$ be the function given by $(g, g')(x) = (g(x), g'(x))$, and let $\mathcal{G}^2 = \{(g, g') : g, g' \in \mathcal{G}\}$. The Natarajan dimension of $\mathcal{G}^2$ is at most $2d$ by Lemma 4. Moreover, consider the loss $c : \mathcal{G}^2 \times \mathcal{X} \to \{0, 1\}$ given by $c(g, g', x) = \mathbb{I}\{g(x) \neq g'(x)\}$. Applying Lemma 2 with the chosen size of $|S'|$ ensures that with probability at least 0.99 every pair $(g, g') \in \mathcal{G}^2$ satisfies

$$\left| \mathbb{E}_{x \sim P}[c(g, g', x)] - \frac{1}{|S'|} \sum_{x \in S'} c(g, g', x) \right| \leq \frac{\gamma}{m}.$$

By the definition of $\hat{\mathcal{G}}$, for every $g \in \mathcal{G}$, there exists $\hat{g} \in \hat{\mathcal{G}}$ for which $c(g, \hat{g}, x) = 0$ for all $x \in S'$, which implies that $\Pr_{x \sim P}(g(x) \neq \hat{g}(x)) \leq \gamma/m$.

Using Lemma 1 to bound the size of $\hat{\mathcal{G}}$, we have that

$$|\hat{\mathcal{G}}| \le |S'|^d |\mathcal{Y}|^{2d} = O\left(\left(\frac{m^2}{\gamma^2} d |\mathcal{Y}|^2 \log |\mathcal{Y}|\right)^d\right).$$

Since this construction succeeds with non-zero probability, we are guaranteed that such a set $\hat{\mathcal{G}}$ exists. Finally, by an identical uniform convergence argument, it follows that if $S$ is a fresh i.i.d. sample of the size given in Item 2 of the lemma's statement, then, with probability at least $1 - \delta$, every $g$ and $\hat{g}$ will disagree on at most a $2\gamma/m$-fraction of $S$, since they disagree with probability at most $\gamma/m$ on $P$.

Next, let $\hat{\mathcal{H}} = \{h_{\vec{g},\eta} : \vec{g} \in \hat{G}^m, \eta \in \hat{\Delta}_m\}$ be the same family as $\mathcal{H}$, except restricted to choosing functions from $\hat{\mathcal{G}}$ and mixing weights from $\hat{\Delta}_m$. Using the size bounds above and the fact that $\binom{N}{m} = O((\frac{N}{m})^m)$, we have that

$$|\hat{\mathcal{H}}| = \binom{|\hat{\mathcal{G}}|}{m} \cdot |\hat{\Delta}_m| = O\left(\frac{(dm^2|\mathcal{Y}|^2 \log(m|\mathcal{Y}|/\gamma))^{dm}}{\gamma^{(2d+1)m}}\right).$$

Suppose that $h$ is the mixture of $g_1, \ldots, g_m \in \mathcal{G}$ with weights $\eta \in \Delta_m$. Let $\hat{g}_i$ be the approximation to $g_i$ for each $i$, let $\hat{\eta} \in \hat{\Delta}_m$ be such that $\|\eta - \hat{\eta}\|_1 \le \gamma$, and let $\hat{h}$ be the random mixture of $\hat{g}_1, \ldots, \hat{g}_m$ with weights $\hat{\eta}$. For an individual $x$ drawn from $P$, we have $g_i(x) \ne \hat{g}_i(x)$ with probability at most $\gamma/m$, and therefore they all agree with probability at least $1 - \gamma$. When this event occurs, we have $\|h(x) - \hat{h}(x)\|_1 \le \|\eta - \hat{\eta}\|_1 \le \gamma$.

The second part of the claim follows by similar reasoning, using the fact that for the given sample size $|S|$, with probability at least $1 - \delta$, every $g \in \mathcal{G}$ disagrees with its approximation $\hat{g} \in \hat{\mathcal{G}}$ on at most a $2\gamma/m$-fraction of $S$. This means that $\hat{g}_i(x) = g_i(x)$ for all $i \in [m]$ on at least a $(1 - 2\gamma)$-fraction of the individuals $x$ in $S$. For these individuals, $\|h(x) - \hat{h}(x)\|_1 \le \|\eta - \hat{\eta}\|_1 \le \gamma$. $\qquad\square$

Combining the generalization guarantee for finite families given in Lemma 5 with the finite approximation given in Lemma 6, we are able to show that envy-freeness also generalizes for $\mathcal{H}(\mathcal{G}, m)$.

*Proof of Theorem 3.* Let $\hat{\mathcal{H}}$ be the finite approximation to $\mathcal{H}$ constructed in Lemma 6. If the sample is of size $|S| = O(\frac{1}{\gamma^2}(dm\log(dm|\mathcal{Y}|\log|\mathcal{Y}|/\gamma) + \log\frac{1}{\delta}))$, we can apply Lemma 5 to this finite family, which implies that for any $\beta' \ge 0$, with probability at least $1 - \delta/2$ every $\hat{h} \in \hat{\mathcal{H}}$ that is $(\alpha', \beta')$-pairwise-EF on $S$ (for any $\alpha'$) is also $(\alpha' + \gamma, \beta')$-EF on $P$. We apply this lemma with $\beta' = \beta + 2\gamma$. Moreover, from Lemma 6, we know that if $|S| = O(\frac{m^2}{\gamma^2}(d\log|\mathcal{Y}| + \log\frac{1}{\delta}))$, then with probability at least $1 - \delta/2$, for every $h \in \mathcal{H}$, there exists $\hat{h} \in \hat{\mathcal{H}}$ satisfying $\|h(x) - \hat{h}(x)\|_1 \le \gamma$ for all but a $2\gamma$-fraction of the individuals in $S$. This implies that on all but at most a $4\gamma$-fraction of the pairs in $S$, $h$ and $\hat{h}$ satisfy this inequality for both individuals in the pair. Assume these high probability events occur. Finally, from Item 1 of the lemma we have that $\Pr_{x_1,x_2 \sim P}(\max_{i=1,2} \|h(x_i) - \hat{h}(x_i)\|_1 > \gamma) \le 2\gamma$.

Now let $h \in \mathcal{H}$ be any classifier that is $(\alpha, \beta)$-pairwise-EF on $S$. Since the utilities are in $[0, 1]$ and $\max_{x=x_i,x_i'} \|h(x) - \hat{h}(x)\|_1 \le \gamma$ for all but a $4\gamma$-fraction of the pairs in $S$, we know that $\hat{h}$ is $(\alpha + 4\gamma, \beta + 2\gamma)$-pairwise-EF on $S$. Applying the envy-freeness generalization guarantee (Lemma 5) for $\hat{\mathcal{H}}$, it follows that $\hat{h}$ is also $(\alpha + 5\gamma, \beta + 2\gamma)$-EF on $P$. Finally, using the fact that

$$\Pr_{x_1,x_2 \sim P}\left(\max_{i=1,2} \|h(x_i) - \hat{h}(x_i)\|_1 > \gamma\right) \le 2\gamma,$$

it follows that $h$ is $(\alpha + 7\gamma, \beta + 4\gamma)$-EF on $P$. $\qquad\square$

It is worth noting that the (exponentially large) approximation $\hat{\mathcal{H}}$ is only used in the generalization analysis; importantly, an ERM algorithm need not construct it.

# D Appendix for Section 5

Here we describe details of the transformation of the optimization problem from (2) to (4). Firstly, softening constraints of (2) with slack variables, we obtain

$$\min_{g_k \in \mathcal{G}, \xi \in \mathbb{R}_{\geq 0}^{n \times n}} \quad \sum_{i=1}^{n} L(x_i, g_k(x_i)) + \lambda \sum_{i \neq j} \xi_{ij}$$

$$\text{s.t.} \quad USF_{ii}^{(k-1)} + \tilde{\eta}_k u(x_i, g_k(x_i)) \geq USF_{ij}^{(k-1)} + \tilde{\eta}_k u(x_i, g_k(x_j)) - \xi_{ij} \quad \forall (i,j).$$

Here, $\xi_{ij}$ basically captures how much $i$ envies $j$ under the selected assignments (note that, $\xi_{ij}$ is 0 if the pair is non-envious, so that the algorithm does not go increasing negative envy at the cost of positive envy for someone else). Plugging in optimal values of the slack variables, we obtain

$$\min_{g_k \in \mathcal{G}} \quad \sum_{i=1}^{n} L(x_i, g_k(x_i))$$

$$+ \lambda \sum_{i \neq j} \max \left( USF_{ij}^{(k-1)} + \tilde{\eta}_k u(x_i, g_k(x_j)) - USF_{ii}^{(k-1)} - \tilde{\eta}_k u(x_i, g_k(x_i)), 0 \right). \quad (8)$$

Next, we perform convex relaxation of different components of this objective function. For this, let's observe the term $L(x_i, g_k(x_i))$. And, let $\vec{w}$ denote the parameters of $g_k$. By definition, we have

$$w_{g_k(x_i)}^\top x_i \geq w_{y'}^\top x_i$$

for any $y' \in \mathcal{Y}$. This implies that

$$L(x_i, g_k(x_i)) \leq L(x_i, g_k(x_i)) + w_{g_k(x_i)}^\top x_i - w_{y'}^\top x_i$$

$$\leq \max_{y \in \mathcal{Y}} \left\{ L(x_i, y) + w_y^\top x_i - w_{y'}^\top x_i \right\},$$

giving us a convex upper bound on the loss $L(x_i, g_k(x_i))$. As this holds for any $y' \in \mathcal{Y}$, we choose $y' = y_i$ as defined in the main body, since it leads to the lowest achievable loss value. Therefore, we have

$$L(x_i, g_k(x_i)) \leq \max_{y \in \mathcal{Y}} \left\{ L(x_i, y) + w_y^\top x_i - w_{y_i}^\top x_i \right\}.$$

This right hand side is basically an upper bound which apart from encouraging $\vec{w}$ to have the highest dot product with $x_i$ at $y_i$, also penalizes if the margin by which this is higher is not enough (where the margin depends on other losses $L(x_i, y)$). This surrogate loss is very similar to multi-class support vector machines. We perform similar relaxations for the other two components of the objective function. In particular, for the $u(x_i, g_k(x_i))$ term, we have

$$-u(x_i, g_k(x_i)) \leq \max_{y \in \mathcal{Y}} \left\{ -u(x_i, y) + w_y^\top x_i - w_{b_i}^\top x_i \right\},$$

where $b_i$ is as defined in the main body. Finally, for the remaining term, we have

$$u(x_i, g_k(x_j)) \leq \max_{y \in \mathcal{Y}} \left\{ u(x_i, y) + w_y^\top x_j - w_{s_i}^\top x_j \right\},$$

where $s_i$ is as defined in the main body[5]. On plugging in the convex surrogates of all three terms in Equation (8), we obtain the optimization problem (4).

## Footnotes

[5]Note that, instead of using $s_i$, an alternative to use in this equation is $b_j$. In particular, for a pair $(i, j)$, using $s_i$ encourages the assignment to give $i$ their favorite outcome while $j$ the outcome that $i$ likes the least (and hence causing $i$ to envy $j$ as less as possible), while using $b_j$ encourages the assignment to give both $i$ and $j$ their favorite outcomes (pushing the assignment to just give everyone their favorite outcomes).