[Reviews · NeurIPS 2019]

Reviewer 1



The paper introduces a novel notion of fairness. This new notion of fairness and its motivation are clearly explained; in particular, the paper provides a clear explanation of the settings in which this notion is applicable, including a honest discussion of when it is NOT applicable (this reviewer wishes more papers in the field did this). This fairness definition is well-motivated and elegant, and it introduces a conceptually novel perspective to ML fairness. To me, it seems likely that this work will have a significant influence on future research in the ML fairness field. The paper also provides a theoretical analysis of envy-free classification, focusing on generalization results. These results are elegant and nontrivial. The paper also includes preliminary results about the computational problem of performing envy-free ERM. The proposed heuristic is nontrivial, but considerably less elegant than the theoretical results. The experimental results give evidence that the proposed heuristic performs reasonably well, but do not compare the proposed methods to any nontrivial baselines (the only comparisons are to the optimal and the random classifiers). The paper is very well written overall, even if sometimes it seems to use more notation than necessary. (E.g., the paragraph starting with "Formally," in Section 4 does not seem to provide much value; the intuitive explanation in words is completely clear and much easier to read than the formal definition.) Overall, this is a very strong paper that should be of interest to the entire ML fairness community.

Reviewer 2



The paper is well written and approaches the problem methodically. This reviewer thought the mathematical formulation well approximates intuitive notions of envy-free classification. That being said, what is missing are concrete suggestions for utility and loss functions. The example provided in the Appendix is useful to motivate randomized classifiers but no example is provided overall to highlight the broader social value of envy-free classification. Theorem 1 is a nice result which initiates the discussion of generalization to the true distribution. Theorem 2 is a very nice result and the covering/probabilistic argument is well explained in the proof. While the reviewer understands that Theorem 2 is provided as an illustrative result, the reviewer suggests the minor adjustment of abstracting the side length rather than fixing it to 1/4 and not fixing the epsilon to 1/10. Thus, upon glancing at the result, a reader could see the dependence between the number of samples, the probabilistic term, and epsilon. As it stands, the result is murky at first glance. The reviewer likes the use of the Natarajan dimension in the way of Theorem 3. It is a good result which points the direction to where future research on this topic should go. The proof of Theorem 3 is technical and well-executed. Lemma 4 extends known results about Natarajan dimension and should be of independent interest. This reviewer has the most concerns about the empirical section of the paper. The authors attempt to illustrate their results on synthetic data so as to reverse engineer the randomizing classifier given the optimal envy free result on linear one-vs-all classifiers but (while serving as a simple example to Theorem 3) it is too contrived to raise concerns about broader applicability. The method appears to yields good results, however, the reviewer takes issue with how the dataset was generated. While the reviewer understands that the model is not as easily tested by using existing datasets, the reviewer would have liked to see a practical result which specifies envy classifiers for a particular family of distributions. Then, apply the algorithm on a dataset which is sampled from this distribution. As it stands, the empirical methodology fixes a particular envy-free classifier and creates a dataset around this classifier. The reviewer feels that this is slightly backward and does not provide a good example of the method in practice. In particular, the following details are opaque: (i) when the authors say, "overall *** is envy-free", do they mean on average? This isn't clear at all and could be clarified right up front even as they argue for randomizing classifiers. (ii) What is most concerning in section 5.3 is, how exactly is envy computed? For every pair and then averaged over all pairs (for the optimal $\eta$?) (iii) In lines 318-320, are the authors referring to $\alpha$ and $\beta$ by providing the fractions? This is entirely unclear. (iv) lastly, figures 3 and 4 could have much more improved legends -- surely, eta can be written as a symbol; the CDF plot isn't very clear either in terms of how it was obtained. Finally, a few more comments: 1) the authors suggest utility could be easily identified in applications but don't do so easily in an example. This reduces the charm of their elegant technical results. minor: in theorem 1, there appears to be an extra bracket following the sup_x\in\mathcal{X} which isn't needed. Nevertheless, on the whole, the paper is well written and the results are compelling.

Reviewer 3



The paper offers an interesting, original notion of envy-free fairness that is well-studied in other disciplines such as sociology, psychology and economics, and applying it to machine learning. The paper is well-written and easy to follow. It offers nice technical results on generalization for simple classifiers given the fairness constraint as (a,b) envy-freeness. While the paper offers a new notion of fairness, it does not discuss in details its justification and relationship with existing studies in other disciplines, but rather taking it as a premise of the paper. It also makes a very strong assumption of the existence of a particular form of utility function u(x, h(x)). Both of the two choices undermine the validity and usefulness of the results in the paper.

[Author Response · NeurIPS 2019]

We thank the reviewers for their helpful feedback. Below we address specific comments.

**Reviewer #1:** The reviewer's point about the clarity of the algorithm in Section 5 is well taken. Please note that some omitted details were relegated to Appendix E. We'd be happy to flesh out the algorithm's description in the body of the paper, as the reviewer suggests.

**Reviewer #2:** The reviewer's main concern has to do with the experimental methodology. We understand the comment that our methodology (generate instances in a way that there is an envy-free classifier with zero loss) is "slightly backward." We wish to clarify, though, that the reasoning behind this experimental design is that, if we generated instances as the reviewer suggests, we wouldn't be able to identify the envy-free classifier that minimizes loss, and, consequently, we wouldn't be able to measure how far our algorithm is from the optimum. Since there are no existing algorithmic benchmarks, we would argue that it's sensible to evaluate our algorithm by generating instances in a way that the optimal solution — i.e., the minimum loss achievable by an envy-free classifier — is known upfront, as we say in lines 281–283.

Let us now answer the reviewer's specific questions, using the given numbering; in our revision we will clarify all issues the reviewer listed.

(i) This is referring to expected utility. Please see lines 131–136. The word "overall" is confusing and will be deleted.

(ii) Exactly, envy is computed for each pair and then averaged over pairs. In Figure 3, "negative envy is replaced with 0, to avoid obfuscating positive envy." Please see lines 310–311.

(iii) Absolutely, these fractions correspond to $\alpha$ and $\beta$ from Definition 1. The purpose here is not to make a technical connection to these parameters, though, but rather to make Figure 4 more concrete by giving examples of two points on the solid orange and magenta lines.

**Reviewer #3:** The reviewer notes that the "paper offers an interesting, original notion of envy-free fairness," that "the paper is well-written," and that "it offers nice technical results." On the negative side, the reviewer raises two issues, regarding the notion of fairness and the existence of utility functions. Since both points are rather terse, we weren't quite sure what specifically the reviewer is concerned about — we apologize if we misunderstood.

*On the notion of envy-freeness:* The reviewer writes that envy-freeness is "well-studied in other disciplines such as sociology, psychology and economics," so it seems that the importance of envy-freeness as a notion of fairnss isn't in question. Rather, if we understand correctly, the reviewer is questioning whether the insights from these other disciplines carry over to the machine learning domain in practice. While human-subject experiments are needed to definitively answer this question,[1] we note that there is a significant body of empirical work on envy-freeness in computational fair division [1], HCI [3], and behavioral economics [2]. The main insight is that people perceive situations where they are envious — i.e., those where they have a higher utility for someone else's outcome than for their own — as unfair; there is no reason why the same conclusion wouldn't hold in the classification setting, as it shares many of the same characteristics. We'd be happy to elaborate on this point in our revision.

*On the existence of utilities:* The reviewer writes that the paper "makes a very strong assumption of the existence of a particular form of utility function $u(x, h(x))$." We actually view this as a very mild assumption. All we're assuming is that each individual has a utility for each outcome, and the utility for a distribution over outcomes is the expected utility. Such utility functions, known as von Neumann-Morgenstern utilities, are the basis for much of the work in economics, decision theory, algorithmic game theory, and related disciplines.

## Footnotes

[1]There are preciously few such studies even with respect to the well established notions of fairness in machine learning.

# References

[1] Y. Gal, M. Mash, A. D. Procaccia, and Y. Zick. Which is the fairest (rent division) of them all? *Journal of the ACM*, 64(6): article 39, 2017.

[2] D. K. Herreiner and C. D. Puppe. Envy freeness in experimental fair division problems. *Theory and decision*, 67(1):65–100, 2009.

[3] M. K. Lee and S. Baykal. Algorithmic mediation in group decisions: Fairness perceptions of algorithmically mediated vs. discussion-based social division. In *Proceedings of the ACM Conference on Computer Supported Cooperative Work and Social Computing (CSCW)*, pages 1035–1048, 2017.



[Meta-Review · NeurIPS 2019]

The main points in favor of this paper are its elegant formulation of envy-freeness as a way to expand the literature on fairness (the idea being that a process is fair if no one envies the decision for any one else) The reviewers acknowledged that the connection to algorithmic fairness and the broader literature on fairness definitions was weak, but appreciated the technical development of the paper itself notwithstanding. The paper not only introduces the new notion but makes good strides on the topic of how to use it - how to construct classifiers that respect this notion and and how they might generalize.